# SARS-CoV-2 nsp16 is regulated by host E3 ubiquitin ligases, UBR5 and MARCHF7

**Li Tian[1†], Zongzheng Zhao[2†], Wenying Gao[1], Zirui Liu[2], Xiao Li[2]\*, Wenyan Zhang[1]\*, Zhaolong Li[1]\***

[1]Department of Infectious Diseases, Infectious Diseases and Pathogen Biology Center, Institute of Virology and AIDS Research, Key Laboratory of Organ Regeneration and Transplantation of The Ministry of Education, The First Hospital of Jilin University, Changchun, China; [2]Research Unit of Key Technologies for Prevention and Control of Virus Zoonoses, Chinese Academy of Medical Sciences, Changchun Veterinary Research Institute, Chinese Academy of Agricultural Sciences, Changchun, China

**\*For correspondence:**
skylee6226@163.com (XL);
zhangwenyan@jlu.edu.cn (WZ);
lizhaolong@jlu.edu.cn (ZL)

[†]These authors contributed equally to this work

## eLife Assessment

This **important** work advances our understanding of how the SARS-CoV-2 Nsp16 protein is regulated by host E3 ligases to promote viral mRNA capping. Support for the overall claims in the revised manuscript is **convincing** . This work will be of interest to those working in host-viral interactions and the role of the ubiquitin-proteasome system in viral replication.

**Abstract** Severe acute respiratory syndrome coronavirus 2 (SARS-CoV-2), the causative agent of coronavirus disease 2019 (COVID-19), remains a global public health threat with considerable economic consequences. The nonstructural protein 16 (nsp16), in complex with nsp10, facilitates the final viral mRNA capping step through its 2′-O-methylase activity, helping the virus to evade host immunity and prevent mRNA degradation. However, nsp16 regulation by host factors remains poorly understood. While various E3 ubiquitin ligases interact with SARS-CoV-2 proteins, their roles in targeting nsp16 for degradation remain unclear. In this study, we demonstrate that nsp16 undergoes ubiquitination and proteasomal degradation mediated by the host E3 ligases UBR5 and MARCHF7. UBR5 induces K48-linked ubiquitination, whereas MARCHF7 promotes K27-linked ubiquitination, independently suppressing SARS-CoV-2 replication in cell cultures and in mice. Notably, UBR5 and MARCHF7 also degrade nsp16 variants from different viral strains, exhibiting broad-spectrum antiviral activity. Our findings reveal novel antiviral mechanisms of the ubiquitin-proteasome system (UPS) and highlight their potential therapeutic targets against COVID-19.

## Introduction

Severe acute respiratory syndrome coronavirus 2 (SARS-CoV-2) remains a global public health threat, with over 777 million coronavirus disease 2019 (COVID-19) cases and 7.1 million deaths as of March 2025 (reported by World Health Organization). The virus encodes 4 structural proteins, 16 nonstructural proteins (nsps) involved in replication and transcription, and several accessory proteins (ORFs) linked to immune evasion (*V'kovski et al., 2021*; *Zhang et al., 2020*). Many viruses, including Ebola (*Valle et al., 2021*), dengue (*Jung et al., 2018*), and reovirus (*Furuichi et al., 1975*), encode 2′-O-methyltransferases (2′-O-MTases) that mimic the host 5′ cap structure to evade innate immune recognition (*Daffis et al., 2010*; *Züst et al., 2011*). Similarly, SARS-CoV-2 nsp16, in complex with nsp10, functions as a 2′-O-MTase, modifying capped viral RNA from 'cap-0' to 'cap-1' (*Benoni et al.,*

*2021*; *Park et al., 2022*). This modification protects the virus from host antiviral defenses, such as MDA5 recognition and IFIT1 restriction (*Bergant et al., 2022*; *Russ et al., 2022*). Beyond immune evasion, nsp16 disrupts global host mRNA splicing, reducing cellular protein and mRNA levels (*Banerjee et al., 2020*). Nsp16 also enhances SARS-CoV-2 entry by upregulating TMPRSS2 expression (*Han et al., 2023*), underscoring its role as a virulence factor. Notably, nsp16-deficient strains exhibit low pathogenicity and elicit robust immune responses, suggesting their potential as live-attenuated vaccines (*Balieiro et al., 2022*). Given the critical functions of nsp16, inhibitors targeting nsp16 or the nsp16-nsp10 complex have been developed (*Klima et al., 2022*; *Nguyen et al., 2023*). Therefore, understanding host factors that regulate nsp16 is essential for identifying new therapeutic strategies.

The ubiquitin-proteasome system (UPS) is a key pathway for targeted protein degradation, often described as 'the molecular kiss of death' (*Tieroyaare Dongdem and Adiyaga Wezena, 2022*; *Park et al., 2020*). Substrate ubiquitination involves a cascade of three enzymes: E1 activating enzyme, E2 conjugating enzyme, and E3 ubiquitin (Ub) ligase (*Zheng and Shabek, 2017*). Some E3 Ub ligases function as monomers, whereas others assemble into multi-subunit complexes. Regardless of their structure, all E3 ligases play a crucial role in substrate recognition, dictating UPS specificity. In certain complexes, an additional protein may function as a substrate receptor, mediating substrate-specific recognition (*Iconomou and Saunders, 2016*; *Yang et al., 2021*; *Li et al., 2021*; *Wang et al., 2022*). Over 600 E3 ligases encoded in the human genome regulate diverse cellular processes in response to various signals (; *Garcia-Barcena et al., 2020*). Dysregulation of their activity is linked to multiple diseases, making them attractive therapeutic targets (*Humphreys et al., 2021*). Based on conserved domains and Ub transfer mechanisms, E3 ligases are categorized into three classes: Really Interesting New Gene (RING), homologous to the E6AP carboxyl terminus (HECT), and RING-between-RING (RBR) ligases (*Garcia-Barcena et al., 2020*; *Morreale and Walden, 2016*). RING E3s transfer Ub directly from the E2-Ub complex to substrates (*Deshaies and Joazeiro, 2009*), whereas HECT-type E3s first attach Ub to their own catalytic cysteine before transferring it (*Huibregtse et al., 1995*). These distinct mechanisms highlight the complexity and versatility of E3 ligase function in cellular regulation (*Walden and Rittinger, 2018*; *Garcia-Barcena et al., 2020*).

To investigate the relationship between SARS-CoV-2 and host factors, we focused on UPS proteins that interact with nsp16. In this study, we demonstrate for the first time that nsp16 undergoes ubiquitination and proteasomal degradation. Notably, two E3 ligases from distinct families—RING-type MARCHF7 and HECT-type UBR5—independently target nsp16 for degradation, disrupting its function and exhibiting strong antiviral activity against SARS-CoV-2. Our findings reveal novel therapeutic targets for combating SARS-CoV-2 and treating COVID-19.

## Results

### SARS-CoV-2 nsp16 is degraded via the Ub-proteasome pathway

Ubiquitination plays a crucial role in SARS-CoV-2 infection and pathogenesis (*Gao et al., 2022*; *Guo et al., 2021*; *Li et al., 2023*; *Zhang et al., 2021*; *Zhang et al., 2024*; *Zhang et al., 2023*). To determine whether SARS-CoV-2 nonstructural proteins (nsps) are UPS-regulated, we examined the effect of the proteasome inhibitor, MG132, on nsps expression. MG132 treatment significantly increased the nsp8, nsp11, and nsp16 abundance in HEK293T cells (*Figure 1A*), suggesting UPS-mediated degradation. Nsp8 undergoes TRIM22-mediated ubiquitination and proteasomal degradation (*Fan et al., 2024*). In this study, we focused on nsp16 for further investigation. To confirm UPS-mediated nsp16 degradation, we tested additional proteasome inhibitors, Bortezomib and Carfilzomib, both of which enhanced nsp16 stability. In contrast, lysosomal inhibitors (Bafilomycin A1 and NH₄Cl) and the autophagy-lysosomal inhibitor, Vinblastine, had no such effect (*Figure 1B*). To further assess the role of the UPS in regulating nsp16 stability, we treated nsp16-expressing cells with cycloheximide, a protein synthesis inhibitor, and measured the half-life of nsp16. MG132 treatment significantly prolonged nsp16 stability, extending its half-life from 2 hr to 15 hr (*Figure 1C and D*).

To identify nsp16-interacting proteins, we performed co-immunoprecipitation (Co-IP) followed by mass spectrometry (MS) analysis, comparing nsp16-expressing cells with and without MG132 treatment (*Figure 1E*). Kyoto Encyclopedia of Genes and Genomes (KEGG) pathway and Gene Ontology (GO) analyses (*Figure 1—figure supplement 1A*) revealed that inhibiting nsp16 degradation increased its interacting proteins, including 43 associated with viral processes, 22 involved in

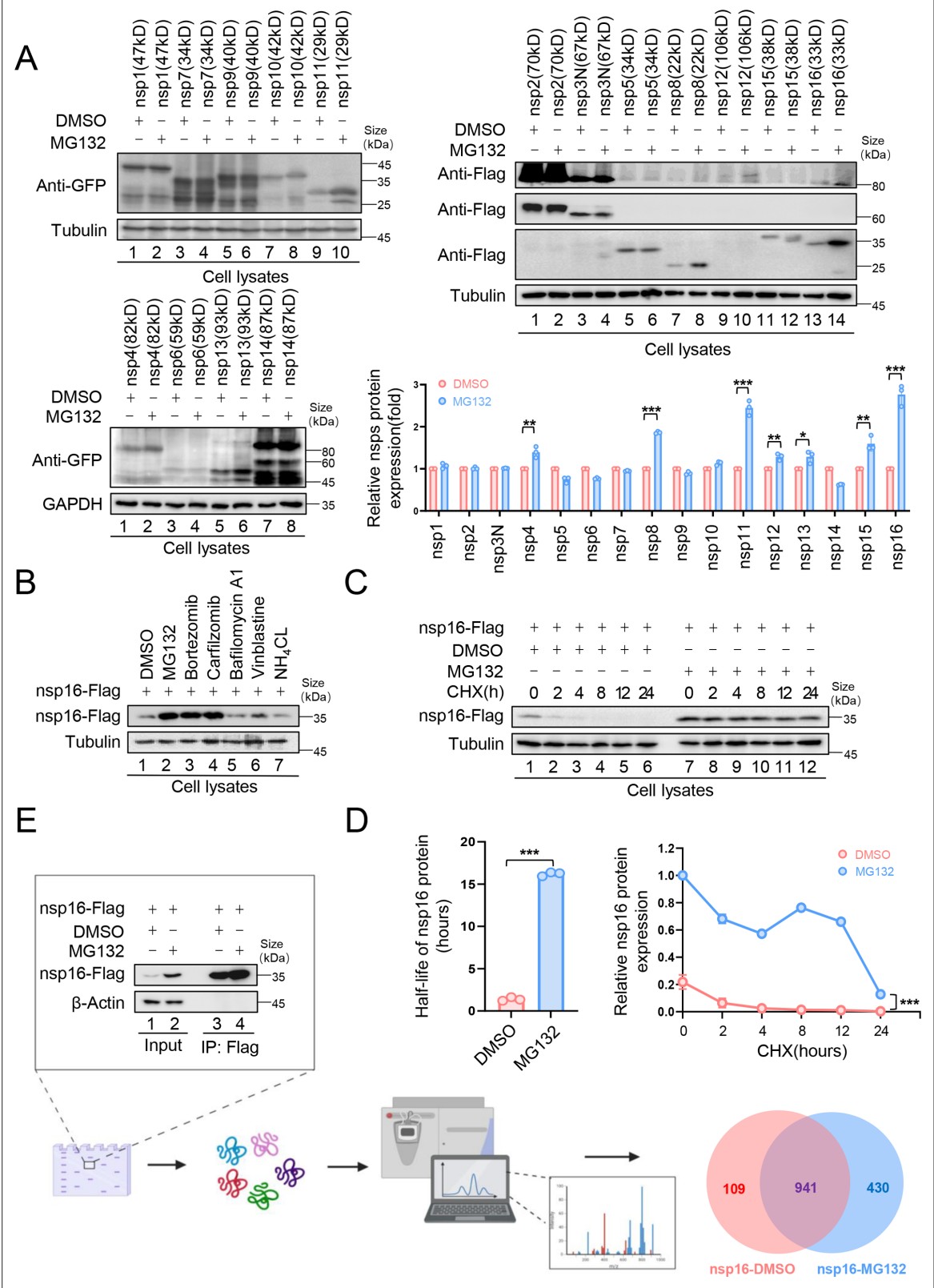

**Figure 1.** The nonstructural protein nsp16 of severe acute respiratory syndrome coronavirus 2 (SARS-CoV-2) was identified that can be degraded through the proteasome pathway. (**A**) The nonstructural proteins nsp8, nsp11, and nsp16 could be restored by the proteasome inhibitor MG132. HEK293T cells in 12-well plates were transfected with the plasmids of 16 nonstructural proteins (nsp1–16) encoded by SARS-CoV-2. Thirty-six hours later, the cells were treated with MG132 (10 µM) or DMSO for 12 hr before collection. The protein level was detected by immunoblotting (IB). Quantification

*Figure 1 continued on next page*

*Figure 1 continued*

of nsp protein levels relative to the control protein is shown. Data are representative of three independent experiments and shown as average ± SD (n=3). Significance was determined by a two-tailed t-test: *p<0.05; **p<0.01; ***p<0.001. (**B**) Proteasomal inhibitors but no other inhibitors stabilized nsp16 protein. HEK293T cells transfected with the nsp16-Flag expression vector were treated with dimethyl sulfoxide (DMSO), MG132 (10 μM), Bortezomib (10 μM), Carfilzomib (10 μM), Bafilomycin A1 (5 μM), Vinblastine (2.5 μM), or NH₄Cl (2.5 μM) for 12 hr prior to harvest. The cell lysates were analyzed by anti-Flag antibody. (**C, D**) The half-life of nsp16 was prolonged by the proteasome inhibitor MG132. (C) HEK293T cells were transfected with the nsp16-Flag-expressing plasmids. 12 hr later, the cells were treated with DMSO or MG132 (10 μM) for 12 hr, then 50 μg/ml cycloheximide (CHX) was added. Cells were harvested at the indicated times to detect the level of viral protein by anti-Flag antibody. (D) Quantification of nsp16 protein levels relative to tubulin at different time points is shown. The half-life of the nsp16 protein was determined based on protein quantification using ImageJ, combined with the protein half-life formula for calculation. Results are shown as mean ± SD (n = 3 independent experiments). ***p<0.001 by a two-tailed t-test. (**E**) Samples were prepared for mass spectrometry, and nsp16 interacting proteins were obtained by immunoprecipitation (IP) (created with BioRender.com and the agreement no. is XR281XWMTN). The plasmids were transfected into HEK293T cells for 48 hr. Treat cells with or without MG132 (10 μM) for 12 hr prior to harvest. The whole-cell lysates were incubated with protein G agarose beads conjugated with anti-Flag antibodies and used for IB with anti-Flag antibodies to detect the nsp16 protein. Samples enriched for proteins were analyzed by mass spectrometry.

The online version of this article includes the following source data and figure supplement(s) for figure 1:

**Source data 1.** PDF file containing original western blots for *Figure 1A, B, C, and E*, indicating the relevant bands and treatments.

**Source data 2.** Original files for western blot analysis displayed in *Figure 1A, B, C, and E*.

**Source data 3.** Numerical data obtained during experiments represented in *Figure 1*.

**Figure supplement 1.** Analysis of mass spectrometry (MS) results.

mRNA stability regulation, and 5 linked to cellular antiviral defense. Additionally, interactions related to mRNA splicing and RNA methylation were identified (*Banerjee et al., 2020*), consistent with the function of nsp16 as a 2'-*O*-MTase. Furthermore, SARS-CoV-2 exploits nsp16 to disrupt host mRNA splicing, which promotes infection (*Park et al., 2022*; *Russ et al., 2022*; *Züst et al., 2011*), further validating our MS data.

As expected, several nsp16-binding proteins were associated with ubiquitination and degradation pathways (*Figure 1—figure supplement 1B*). We identified 4 deubiquitinases (DUBs), the E2 ligase UBE2D3, and 14 proteasomal enzymes. Among them, 6 E3 ligases—UBR5, MARCHF7, HECTD1, TRIM32, MYCBP2, and TRIM21—were selected for further investigation.

## E3 ligases, UBR5 and MARCHF7, independently mediate nsp16 degradation

To identify the E3 ligases responsible for nsp16 degradation, we designed small interfering RNAs (siRNAs) targeting six candidate E3 ligases and assessed their effects on nsp16 abundance. *UBR5* and *MARCHF7* knockdown significantly stabilized nsp16 protein levels (*Figure 2A*). To further investigate their roles in nsp16 degradation, we generated stable cell lines with *UBR5* or *MARCHF7* knockdowns and confirmed silencing efficiency (*Figure 2B*). To determine whether UBR5 and MARCHF7 function cooperatively, we performed dual knockdowns. Silencing *UBR5* in *MARCHF7* knockdown cells, and vice versa, further enhanced nsp16 stability, indicating that these ligases act independently (*Figure 2C*). Overexpression studies further supported their independent roles—MARCHF7 overexpression induced nsp16 degradation even in *MARCHF7* knockdown and *MARCHF7/UBR5* double-knockdown cells. Similarly, UBR5 overexpression promoted nsp16 degradation despite *MARCHF7* knockdown. Silencing efficiencies were validated across all experiments (*Figure 2—figure supplement 1A and B*). Notably, wild-type MARCHF7 and UBR5, but not their mutants lacking functional RING (1–542 aa) or HECT domains, respectively, effectively degraded nsp16 in knockdown cells (*Figure 2D and E*). These findings confirm that UBR5 and MARCHF7 independently mediate nsp16 ubiquitination via their HECT and RING domains, targeting it for proteasomal degradation.

## UBR5 and MARCHF7 mediate distinct Ub linkages on nsp16

We first confirmed nsp16 ubiquitination both with and without exogenous Ub expression (*Figure 3A and B*). To investigate the role of E3 ligases in this process, we examined the impact of *UBR5* and *MARCHF7* knockdown on nsp16 ubiquitination. Silencing either ligase reduced nsp16 ubiquitination compared to the negative control, indicating their involvement in this modification (*Figure 3C*). Ub contains seven lysine residues (K6, K11, K27, K29, K33, K48, and K63), which form polyubiquitin chains with distinct functions that determine the fate of the attached protein (*Grice and Nathan, 2016*). To

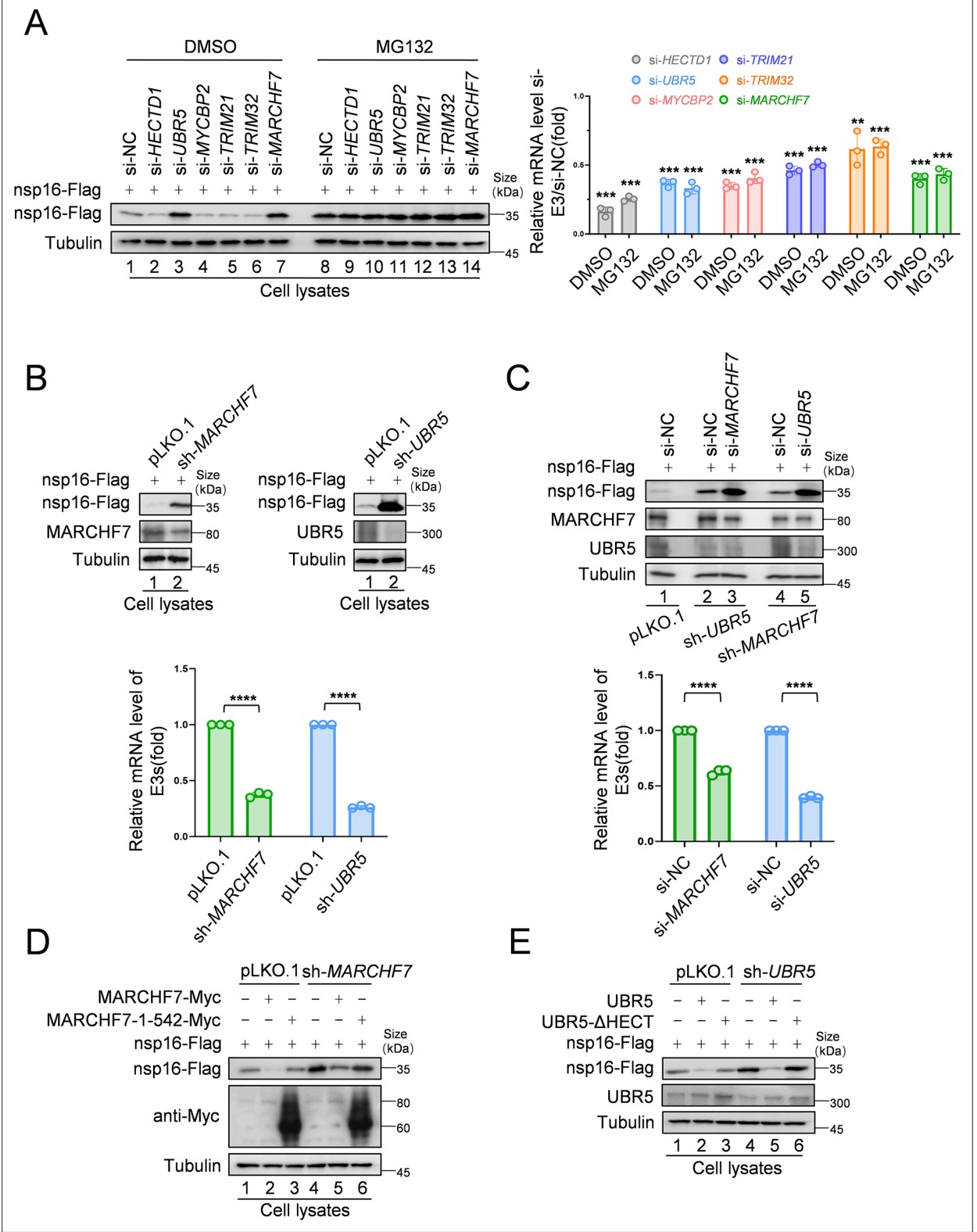

**Figure 2.** MARCHF7 and UBR5 identified as E3 ubiquitin ligases involved in nsp16 protein degradation. (**A**) Knockdown of *MARCHF7* or *UBR5* resulted in nsp16 restoration. HEK293T cells were transfected with small interfering RNA (siRNA) of E3 ligase candidates for 24 hr, followed by co-incubation with the nsp16-Flag-expressing plasmids for 48 hr, treated with MG132 (10 µM) for 16 hr before harvesting, lysed, and subjected to immunoblotting (IB) assay using anti-Flag antibody. RT-qPCR was conducted to determine the mRNA expression levels of E3 ligase candidates. The siRNA targeting regions for the candidate E3 ubiquitin ligase proteins and the targeted regions for RT-qPCR are shown in *Figure 2—figure supplement 1C*. Data are

*Figure 2 continued on next page*

*Figure 2 continued*

representative of three independent experiments and shown as average ± SD (n=3). Significance was determined by a two-tailed t-test: ***p<0.001. (**B**) RNA levels of UBR5 or MARCHF7 from HEK293T cells infected with lentivirus containing control or shRNA targeting *UBR5* or *MARCHF7* for 48 hr and screened with antibiotics for 48 hr. Knockdown cell lines were transfected with plasmids expressing nsp16-Flag, collected at the indicated times, and the protein levels of nsp16, MARCHF7, and UBR5 were detected by IB. (**C**) MARCHF7 and UBR5 acted separately and did not depend on each other. HEK293T cells stably expressing *UBR5* shRNA or *MARCHF7* shRNA were transfected with siRNA of *MARCHF7* or *UBR5* for 24 hr, respectively, followed by co-incubation with the nsp16-Flag-expressing plasmids for 48 hr. The protein levels and the RNA levels of nsp16, UBR5, and MARCHF7 were measured by IB and RT-qPCR, respectively. Data are representative of three independent experiments and shown as average ± SD (n=3). Significance was determined by a two-tailed t-test: ***p<0.001. (**D, E**) In HEK293T cells stably expressing *UBR5* shRNA or *MARCHF7* shRNA, nsp16 was degraded by overexpressed UBR5 or MARCHF7, respectively, whereas the mutant failed to degrade nsp16. The cell lysates were analyzed by anti-Flag antibody.

The online version of this article includes the following source data and figure supplement(s) for figure 2:

**Source data 1.** PDF file containing original western blots for *Figure 2A–E*, indicating the relevant bands and treatments.

**Source data 2.** Original files for western blot analysis displayed in *Figure 2A–E*.

**Source data 3.** Numerical data obtained during experiments represented in *Figure 2*.

**Figure supplement 1.** MARCHF7 and UBR5 degrade nsp16 independently.

**Figure supplement 1—source data 1.** PDF file containing original western blots for *Figure 2—figure supplement 1A and B*, indicating the relevant bands and treatments.

**Figure supplement 1—source data 2.** Original files for western blot analysis displayed in *Figure 2—figure supplement 1A and B*.

**Figure supplement 1—source data 3.** Numerical data obtained during experiments represented in *Figure 2—figure supplement 1*.

---

identify the specific polyubiquitin chain types formed on nsp16 by UBR5 and MARCHF7, we used a series of Ub mutants, each retaining only a single-lysine residue. Except for K33, all single-lysine Ub mutants supported nsp16 ubiquitination to varying degrees, suggesting a complex Ub architecture potentially regulated by multiple E3 ligases or E2-E3 pairs (*Figure 3D*). Further analysis of lysine-specific linkages revealed that MARCHF7 primarily mediates K27-linked ubiquitination of nsp16, whereas UBR5 facilitates K48-linked ubiquitination (*Figure 3E and F*). These findings establish that UBR5 and MARCHF7 independently regulate nsp16 degradation through distinct Ub linkages.

## UBR5 and MARCHF7 directly interact and colocalize with nsp16 in the ER

MS analysis suggested interactions between nsp16 and both UBR5 and MARCHF7. Co-IP experiments confirmed that Myc-tagged MARCHF7 and endogenous UBR5 bind to nsp16 (*Figure 4—figure supplement 1A and B*). To assess whether these interactions are direct, we performed fluorescence resonance energy transfer (FRET) assays. When nsp16-YFP was bleached, the fluorescence intensity of CFP-UBR5 and CFP-MARCHF7 increased, confirming direct interactions (*Figure 4—figure supplement 1C*). ImageJ software was used to quantify relative fluorescence intensities (*Figure 4—figure supplement 1D*). To determine whether UBR5 and MARCHF7 depend on each other for nsp16 binding, we examined the effect of *MARCHF7* knockdown on UBR5-nsp16 interactions, and vice versa, using Co-IP. The interactions remained unchanged, indicating that UBR5 and MARCHF7 independently bind to nsp16 (*Figure 4A and B*). Nsp16 localizes to both the nucleus and cytoplasm (*Zhang et al., 2020*). UBR5 (containing two nuclear localization signals) and MARCHF7 are also present in both compartments (*Muñoz-Escobar et al., 2015*; *Shearer et al., 2018*; *Nathan et al., 2008*). Immunofluorescence (IF) staining revealed that UBR5 and MARCHF7 colocalized with nsp16 predominantly in the cytoplasm, with some nuclear colocalization observed in HeLa cells (*Figure 4—figure supplement 1E*). Similar results were obtained in nsp16-transfected HEK293T cells using antibodies against endogenous UBR5 and MARCHF7 (*Figure 4—figure supplement 1F*).

To pinpoint the specific cellular compartment where these interactions occur, we co-transfected UBR5-CFP or MARCHF7-CFP with nsp16-YFP and performed immunostaining using organelle-specific markers: COX5A (mitochondria), PDI (ER), and GM130 (Golgi apparatus). Notably, UBR5 and MARCHF7 both interacted with nsp16 and colocalized with PDI, but not with COX5A or GM130 (*Figure 4C and D*), indicating that both E3 ligases interact with nsp16 primarily in the ER.

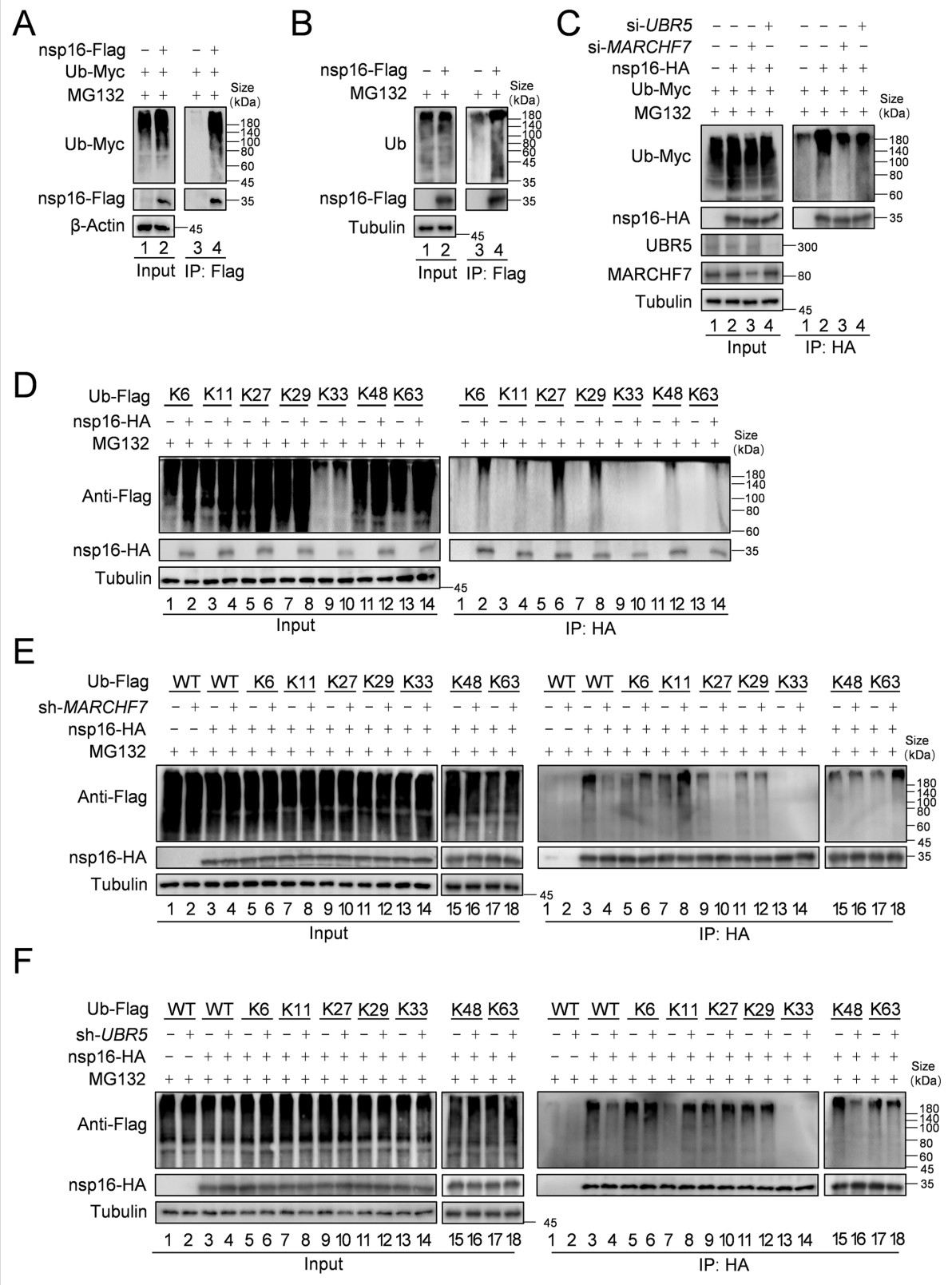

**Figure 3.** MARCHF7 or UBR5 catalyze the formation of K27-type or K48-type ubiquitin chains of nsp16, respectively. (**A**) Nsp16 can be ubiquitinated. HEK293T cells co-transfected with ubiquitin-Myc and nsp16-Flag or transfected with nsp16-Flag alone. The cells were treated with MG132 for 12 hr before collection. The whole-cell lysates were incubated with anti-Flag beads and used for immunoblotting (IB) with anti-Myc or anti-Flag antibodies to detect the polyubiquitination chain of nsp16. (**B**) Assess the endogenous ubiquitination level of nsp16 protein. Cells were transfected with nsp16-Flag

*Figure 3 continued on next page*

*Figure 3 continued*

or an empty vector and collected 48 hr later. Prior to harvesting, cells were treated with MG132 for 16 hr. Co-immunoprecipitation (Co-IP) experiments were then performed to analyze the endogenous ubiquitination level of nsp16. (**C**) The level of ubiquitination of nsp16 decreased with decreasing the protein levels of MARCHF7 or UBR5. E3 was knocked down by transfection with small interfering RNA (siRNA) targeting *UBR5* or *MARCHF7*, and 24 hr later, ubiquitin-Myc and nsp16-HA were co-transfected or nsp16-HA alone. Cells were treated with MG132 for 16 hr before collection. Whole-cell lysates were incubated with anti-HA beads, and polyubiquitinated chains of nsp16 were detected by IB with anti-Myc or anti-HA antibodies. (**D**) Nsp16 can be modified by a variety of ubiquitin chains. HEK293T cells were transfected with either nsp16-HA alone or together with plasmids encoding various mutants of ubiquitin (K6 only, K11 only, K27 only, K29 only, K33 only, K48 only, K63 only). Thirty-six hours later, cells were treated with MG132 for 12 hr. Cell lysates were then subjected to immunoprecipitation, followed by IB to analysis. (**E, F**) MARCHF7 or UBR5 causes nsp16 to be modified by the K27-type or K48-type ubiquitin chain. 293T cell lines with or without *MARCHF7* or *UBR5* knockdown were co-transfected with plasmids encoding ubiquitin-WT or various mutants of ubiquitin (K6 only, K11 only, K27 only, K29 only, K33 only, K48 only, K63 only). The other experimental methods were the same as C.

The online version of this article includes the following source data for figure 3:

**Source data 1.** PDF file containing original western blots for *Figure 3A–F*, indicating the relevant bands and treatments.

**Source data 2.** Original files for western blot analysis displayed in *Figure 3A–F*.

## Functional domains of UBR5 and MARCHF7 are required for nsp16 interaction and ubiquitination

UBR5 is a four-domain E3 ligase containing two nuclear localization signals. Its domains include UBA, UBR, PABC, and HECT (*Muñoz-Escobar et al., 2015*; *Figure 4—figure supplement 2A*). Notably, the HECT domain is critical for its E3 ligase activity, as UBR5 must first conjugate to Ub before transferring it to substrates (*Kim et al., 2021*). To determine which UBR5 domain mediates nsp16 ubiquitination, we used UBR5 mutants with individual domain inactivation. Only the HECT domain mutant failed to degrade nsp16 (*Figure 4—figure supplement 2B*), consistent with previous studies (*Zhou et al., 2022*). This mutant also lost the ability to ubiquitinate nsp16, confirming the essential role of the HECT domain in this process (*Figure 4—figure supplement 2C*).

To identify the MARCHF7 region responsible for nsp16 degradation, we generated a series of truncation mutants as described previously (*Figure 4—figure supplement 2D*; *Zhao et al., 2018*; *Nathan et al., 2008*). All mutants lost the ability to degrade nsp16 (*Figure 4—figure supplement 2E*). Further analysis revealed that only the mutant containing the intact N-terminal region (aa 1–542) strongly bound to nsp16, while those retaining the active RING domain region (aa 543–616) did not. Consistently, only wild-type MARCHF7 promoted K27-linked ubiquitination of nsp16 (*Figure 4—figure supplement 2F*). These results indicate that the N-terminal region of MARCHF7 is essential for nsp16 binding, whereas its RING domain is required for K27-linked ubiquitination activity.

## UBR5 and MARCHF7 suppress SARS-CoV-2 replication by targeting nsp16 for degradation

To assess the impact of nsp16 degradation on viral replication, we used a biosafety level-2 (BSL-2) system to generate SARS-CoV-2 transmissible virus-like particles capable of infecting Caco2-N$^{int}$ cells (*Ju et al., 2021*). Knockdown of either *UBR5* or *MARCHF7* significantly increased SARS-CoV-2 replication, with simultaneous knockdown further enhancing viral replication (*Figure 5—figure supplement 1A and B*).

To investigate nsp16 modifications following infection, we infected HEK293T-ACE2 cells with the IME-BJ01 strain for 48 hr. Co-IP using an anti-nsp16 antibody confirmed its interaction with endogenous MARCHF7 and UBR5, along with ubiquitination (*Figure 5A*). In a biosafety level-3 (BSL-3) facility, we further validated the antiviral roles of UBR5 and MARCHF7 using the IME-BJ01 (MOI 0.01) and Omicron strains (MOI 0.001). Knockdown of either E3 ligase in Caco2 cells significantly increased intracellular and secreted viral mRNA levels (M and E genes) for both strains. Viral titers in the supernatants also increased, with *UBR5* knockdown causing over a 1-log rise for the IME-BJ01 strain and even greater effects for Omicron (*Figure 5C–E and F–H* for the IME-BJ01 and Omicron strains, respectively). Immunoblotting (IB) confirmed increased intracellular and secreted N protein levels upon *UBR5* or *MARCHF7* knockdown (*Figure 5I*), with knockdown efficiencies verified (*Figure 5B*). Due to low transfection efficiency in Caco2 cells, we overexpressed UBR5-Myc or MARCH7-Myc in HEK293T-ACE2 cells. Overexpression significantly reduced viral mRNA (M and E genes) and N protein

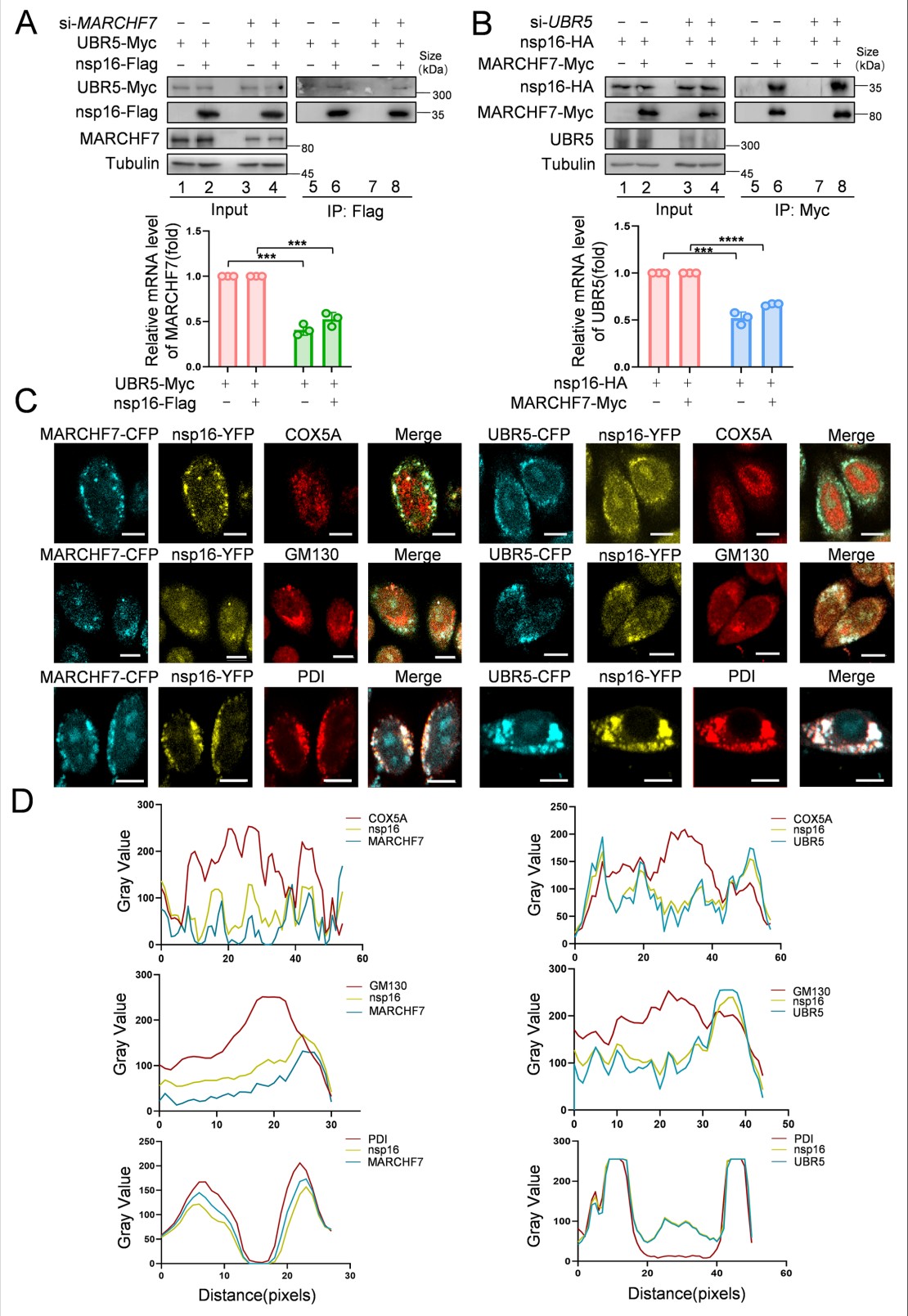

**Figure 4.** MARCHF7 and UBR5 directly interact with nsp16, respectively. (**A, B**) The binding of MARCHF7 or UBR5 to nsp16 was not mutually dependent. The binding of nsp16 to UBR5 or MARCHF7 was identified by co-immunoprecipitation in HEK293T cells transfected into si*MARCHF7* or si*UBR5*, respectively. The immunoprecipitates and input were analyzed by immunoblotting (IB). The knockdown efficiency was detected by RT-qPCR and IB. Data are representative of three independent experiments and shown as average ± SD (n=3). Significance was determined by a two-tailed t-test:

*Figure 4 continued on next page*

*Figure 4 continued*

***p<0.001. (**C, D**) MARCHF7 or UBR5 colocalized with nsp16 in the endoplasmic reticulum. Hela cells were co-transfected with YFP-nsp16 (yellow) and CFP-UBR5 (cyan) or CFP-MARCHF7 (cyan). The organelles were labeled with antibodies against marker proteins of endoplasmic reticulum, Golgi apparatus, and mitochondria respectively (red). The cells were analyzed by confocal microscopy (**C**). Scale bars, 20 µm. The ratio of colocalization was quantified by measuring the fluorescence intensities using ImageJ (**D**).

The online version of this article includes the following source data and figure supplement(s) for figure 4:

**Source data 1.** PDF file containing original western blots for *Figure 4A and B*, indicating the relevant bands and treatments.

**Source data 2.** Original files for western blot analysis displayed in *Figure 4A and B*.

**Source data 3.** Numerical data obtained during experiments represented in *Figure 4*.

**Figure supplement 1.** Interaction of MARCHF7 or UBR5 with nsp16.

**Figure supplement 1—source data 1.** PDF file containing original western blots for *Figure 4—figure supplement 1A and B*, indicating the relevant bands and treatments.

**Figure supplement 1—source data 2.** Original files for western blot analysis displayed in *Figure 4—figure supplement 1A and B*.

**Figure supplement 1—source data 3.** Numerical data obtained during experiments represented in *Figure 4—figure supplement 1*.

**Figure supplement 2.** Domains in which MARCHF7 or UBR5 functions.

**Figure supplement 2—source data 1.** PDF file containing original western blots for *Figure 4—figure supplement 2B, C, E, and F*, indicating the relevant bands and treatments.

**Figure supplement 2—source data 2.** Original files for western blot analysis displayed in *Figure 4—figure supplement 2B, C, E, and F*.

levels in both strains, accompanied by a 0.5-log decrease in viral titers. However, co-transfection with increasing nsp16 levels counteracted these inhibitory effects (*Figure 6A–H*, *Figure 6—figure supplement 1A–H*). To confirm the role of UBR5 and MARCHF7 enzymatic activity, we overexpressed catalytically inactive mutants (UBR5-ΔHECT or MARCHF7-aa 1–542) alongside nsp16. These mutants failed to suppress viral replication. Additionally, gradual increases in nsp16 expression did not further enhance viral replication, with only a slight increase in M mRNA, while E and N protein levels remained unchanged (*Figure 6—figure supplement 2A–H*). These findings demonstrate that UBR5 and MARCHF7 exert antiviral effects by ubiquitinating and degrading nsp16, thereby significantly inhibiting SARS-CoV-2 replication.

## UBR5 and MARCHF7 mediate nsp16 variant degradation

With the continuous emergence of SARS-CoV-2 variants, broad-spectrum antiviral strategies have become essential. To assess whether UBR5 and MARCHF7 mediate various nsp16 variants, we analyzed nsp16 amino acid sequences from the National Center for Biotechnology Information (NCBI) (*Figure 6—figure supplement 3A*). Most variants had one or two mutations, except for XBB.1.9.1, XBB.1.5, and XBB.1.16, which contained more mutations, suggesting a conserved sequence. We used mutagenesis to synthesize nsp16 sequences for these three variants, along with several single-site mutants from other variants. Treatment with MG132 restored nsp16 protein levels for all variants (*Figure 6—figure supplement 3B*), confirming UPS-mediated degradation. Furthermore, *UBR5* or *MARCHF7* knockdown stabilized nsp16 proteins from these variants, though to varying degrees (*Figure 6—figure supplement 3C*). These results suggest that UBR5 and MARCHF7 contribute to broad-spectrum antiviral activity by targeting nsp16 from diverse SARS-CoV-2 variants for degradation.

## SARS-CoV-2 infection reduces UBR5 and MARCHF7 expression

To investigate the impact of SARS-CoV-2 infection on UBR5 and MARCHF7 expression, we analyzed their mRNA and protein levels following infection with the IME-BJ01 and Omicron strains at varying MOIs using RT-qPCR and IB. MARCHF7 expression decreased consistently with increasing viral titers, whereas UBR5 levels initially rose at low titers before declining as infection progressed (*Figure 6—figure supplement 4A–C*). This transient UBR5 upregulation may be linked to its role in interferon-γ-mediated pathways, as previously reported (*Wu et al., 2022*). To confirm these findings in vivo, we examined UBR5 and MARCHF7 mRNA levels in peripheral blood mononuclear cells (PBMCs) from SARS-CoV-2-infected patients with varying disease severity. UBR5 expression negatively correlated with disease progression, whereas MARCHF7 levels showed no significant correlation (*Figure 6—figure*

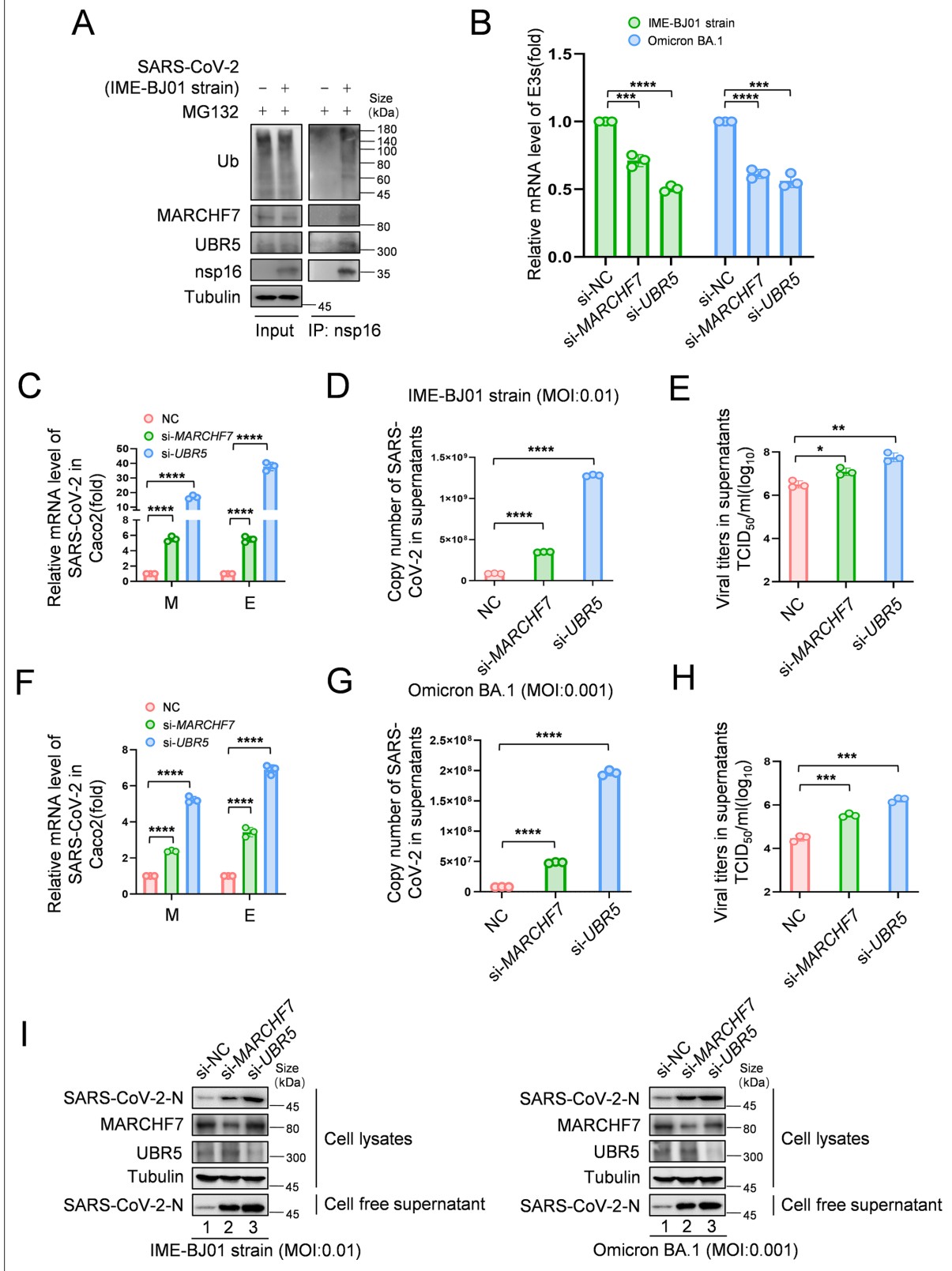

**Figure 5.** Knockdown of *MARCHF7* or *UBR5* promotes viral replication. (**A**) The virus-encoded nsp16 protein interacts with endogenous MARCHF7 and UBR5 and undergoes ubiquitination modification. In 293T-ACE2 cells, with or without IME-BJ01 strain infection (MOI: 0.01), the medium was changed 2 hr post-infection, and cells were harvested 48 hr later, with MG132 treatment added 16 hr before harvesting. nsp16 protein was enriched using protein G beads coupled with the nsp16 antibody, and interactions and ubiquitination were analyzed by immunoblotting (IB) with endogenous antibodies

*Figure 5 continued on next page*

*Figure 5 continued*

against MARCHF7, UBR5, and ubiquitination. (**B–I**) *MARCHF7* and *UBR5* were knocked down by small interfering RNA (siRNA) in Caco2 cells. 24 hr after transfection, the cells were infected with IME-BJ01 strain (MOI: 0.01) (**C–E**) or Omicron BA.1 strain (MOI: 0.001) (**F–H**), respectively. 2 hr post-infection, the supernatant was discarded, and the cells were cultured in Dulbecco's modified Eagle's medium (DMEM) containing 3% fetal bovine serum for 48 hr. The mRNA levels of severe acute respiratory syndrome coronavirus 2 (SARS-CoV-2) M and E genes in the cells (**C, F**) and E genes in supernatant (**D, G**) were detected by RT-qPCR, and the viral titers in supernatant (**E, H**) were measured. The N protein levels of IME-BJ01 or Omicron viruses were detected by IB (**I**). Knockdown efficiencies of *MARCHF7* and *UBR5* were detected by RT-qPCR or IB (**B, I**). Data are representative of three independent experiments and shown as average ± SD (n=3). Significance was determined by one-way ANOVA, followed by a Tukey's multiple comparisons posttest: *p<0.05; **p<0.01; ***p<0.001.

The online version of this article includes the following source data and figure supplement(s) for figure 5:

**Source data 1.** PDF file containing original western blots for *Figure 5A and I*, indicating the relevant bands and treatments.

**Source data 2.** Original files for western blot analysis displayed in *Figure 5A and I*.

**Source data 3.** Numerical data obtained during experiments represented in *Figure 5*.

**Figure supplement 1.** Effect of MARCHF7 or UBR5 on severe acute respiratory syndrome coronavirus 2 (SARS-CoV-2) trVLP.

**Figure supplement 1—source data 1.** Numerical data obtained during experiments represented in *Figure 5—figure supplement 1*.

*supplement 4D*). These results suggest that SARS-CoV-2 may actively suppress host antiviral defenses by downregulating UBR5 and MARCHF7 expression.

## UBR5 and MARCHF7 protect mice from SARS-CoV-2 challenge

Since no specific activators or inhibitors for E3 ligases are available, we used a transient overexpression method using high-pressure tail vein injection of plasmids in mice (*Bonamassa et al., 2011*). To evaluate their antiviral effects in vivo, mice were injected with plasmids encoding UBR5 or MARCHF7, followed by SARS-CoV-2 challenge (*Figure 7A*). E3 ligase overexpression significantly reduced viral E gene copy numbers in the lungs and markedly decreased viral titers (*Figure 7C and D*). Additionally, treated mice experienced less weight loss compared to controls (*Figure 7E*). Histopathological analysis at 5 days post-infection showed that lungs from UBR5- or MARCHF7-treated mice exhibited reduced alveolar contraction and pulmonary edema compared to the severe lesions observed in control mice (*Figure 7F*). IB further confirmed lower N protein abundance in treated mice (*Figure 7G*). These results suggest that UBR5 and MARCHF7 overexpression suppresses SARS-CoV-2 virulence in vivo by promoting nsp16 degradation.

SARS-CoV-2 infection can induce a severe cytokine storm, contributing to high mortality rates (*Song et al., 2020*; *Hojyo et al., 2020*). To assess the impact of UBR5 and MARCHF7 on inflammatory responses, we measured key cytokine levels—including interleukin-6, interleukin-1 receptor antagonist, and interleukin-1β—in the spleens of infected mice (*Makaremi et al., 2022*; *Hu et al., 2021*). Consistent with their antiviral activity, UBR5 and MARCHF7 treatment significantly reduced the production of these inflammatory cytokines (*Figure 7H*).

## Discussion

SARS-CoV-2 nsp16 functions as a 2′-*O*-MTase, catalyzing penultimate nucleotide methylation of the viral RNA cap. This modification allows SARS-CoV-2 to mimic host mRNA, evading immune detection and response (*Lin et al., 2020*; *Balieiro et al., 2022*; *Russ et al., 2022*). High-resolution structures of nsp16 and nsp16-nsp10 heterodimers have been characterized (*Rosas-Lemus et al., 2020*; *Klima et al., 2022*; *Lugari et al., 2010*), providing a foundation for antiviral drug development targeting these complexes (*Balieiro et al., 2022*; *Melo-Filho et al., 2022*). However, host factors that interact with and regulate nsp16 remain largely unknown.

Previous studies, including work from our group, have highlighted the role of the UPS in modulating SARS-CoV-2 infection (*Xu et al., 2022*). Several host E3 ligases—such as RNF5, Cullin4-DDB1-PRPF19, ZNF598, and TRIM7—ubiquitinate and degrade key SARS-CoV-2 proteins, thereby suppressing viral replication (*Li et al., 2023*; *Liang et al., 2022*; *Maimaitiyiming et al., 2022*; *Zhang et al., 2024*). In contrast, SARS-CoV-2 uses DUBs to counteract these defenses and enhance viral replication (*Chen et al., 2024*; *Gao et al., 2024*; *Gao et al., 2022*; *Guo et al., 2021*). In this study, we identify nsp16 as a short-lived protein with a half-life of only 2–3 hr, which is targeted for ubiquitination

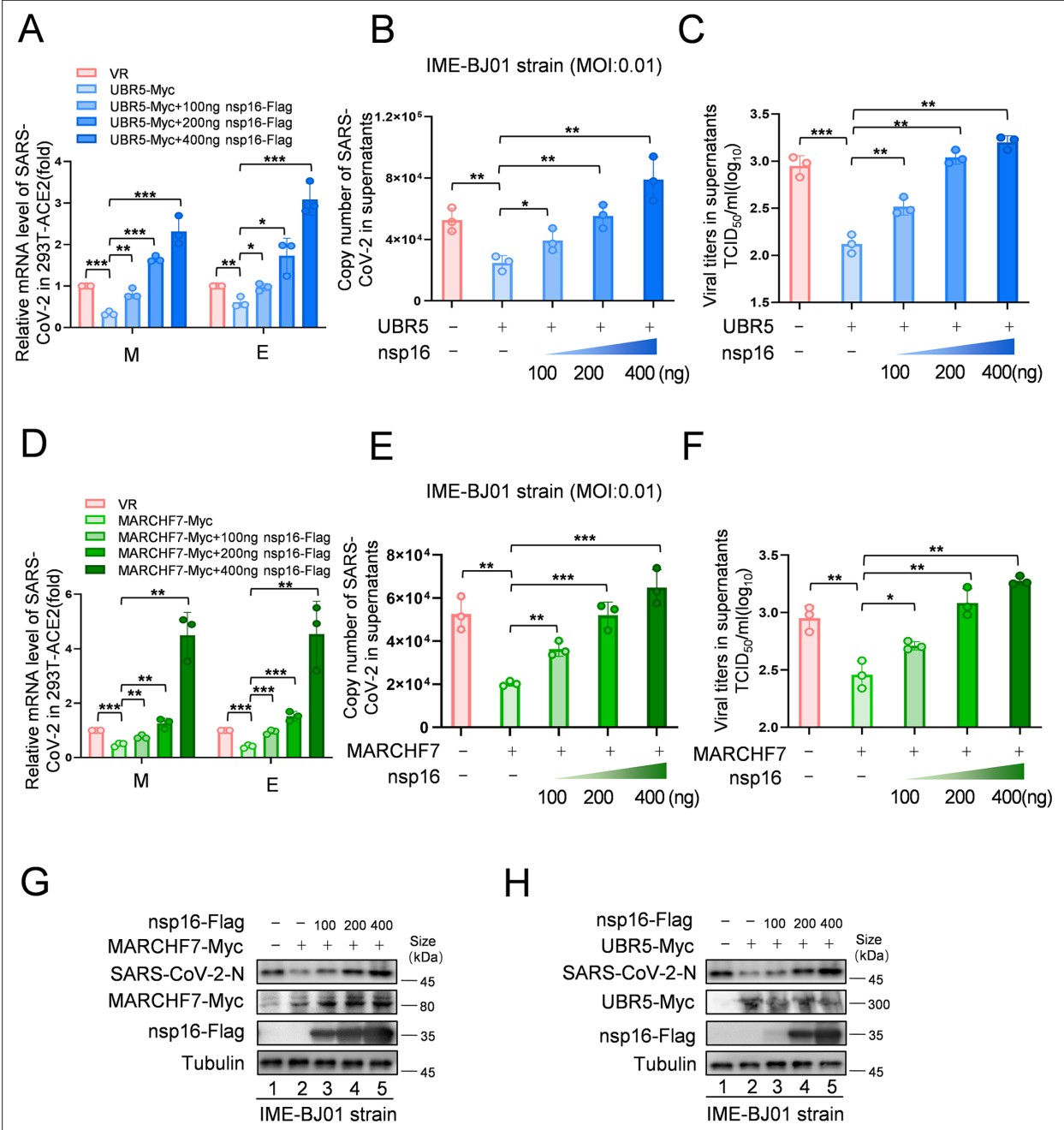

**Figure 6.** Increased levels of nsp16 rescued viral inhibition by UBR5 or MARCHF7. (**A–H**) UBR5 or MARCHF7 was transfected in 293T cells stably overexpressed with ACE2, and the increased doses of nsp16-Flag were transfected simultaneously. After 24 hr, the cells were infected with IME-BJ01 strains. The mRNA levels of M and E genes of the IME-BJ01 strain in the cells (**A, D**) and E gene in supernatant (**B, E**) were detected by RT-qPCR, as well as the detection of viral titers in supernatant (**C, F**). The N protein of the virus and the overexpression efficiency were detected by IB (**G, H**). Data are representative of three independent experiments and shown as average ± SD (n=3). Significance was determined by one-way ANOVA, followed by a Tukey's multiple comparisons posttest. p>0.05; **p<0.01; ***p<0.001. *Figure 6—figure supplement 1* shows data related to infection with Omicron BA.1.

The online version of this article includes the following source data and figure supplement(s) for figure 6:

**Source data 1.** PDF file containing original western blots for *Figure 6G and H*, indicating the relevant bands and treatments.

**Source data 2.** Original files for western blot analysis displayed in *Figure 6G and H*.

**Source data 3.** Numerical data obtained during experiments represented in *Figure 6*.

**Figure supplement 1.** Effect of MARCHF7 or UBR5 on Omicron BA.1 infectivity.

*Figure 6 continued on next page*

*Figure 6 continued*

**Figure supplement 1—source data 1.** PDF file containing original western blots for *Figure 6—figure supplement 1G and H*, indicating the relevant bands and treatments.

**Figure supplement 1—source data 2.** Original files for western blot analysis displayed in *Figure 6—figure supplement 1G and H*.

**Figure supplement 1—source data 3.** Numerical data obtained during experiments represented in *Figure 6—figure supplement 1*.

**Figure supplement 2.** The enzyme activity-deficient mutants do not exhibit antiviral activity, and overexpression of nsp16 does not promote viral replication.

**Figure supplement 2—source data 1.** PDF file containing original western blots for *Figure 6—figure supplement 2G and H*, indicating the relevant bands and treatments.

**Figure supplement 2—source data 2.** Original files for western blot analysis displayed in *Figure 6—figure supplement 2G and H*.

**Figure supplement 2—source data 3.** Numerical data obtained during experiments represented in *Figure 6—figure supplement 2*.

**Figure supplement 3.** MARCHF7 or UBR5 has effects on the mutant of nsp16 in different subtypes of severe acute respiratory syndrome coronavirus 2 (SARS-CoV-2).

**Figure supplement 3—source data 1.** PDF file containing original western blots for *Figure 6—figure supplement 3B and C*, indicating the relevant bands and treatments.

**Figure supplement 3—source data 2.** Original files for western blot analysis displayed in *Figure 6—figure supplement 3B and C*.

**Figure supplement 4.** The expression level of MARCHF7 was negatively correlated with the viral titer, while the expression level of UBR5 was increased at low titer and decreased at high titer.

**Figure supplement 4—source data 1.** PDF file containing original western blots for *Figure 6—figure supplement 4C*, indicating the relevant bands and treatments.

**Figure supplement 4—source data 2.** Original files for western blot analysis displayed in *Figure 6—figure supplement 4C*.

**Figure supplement 4—source data 3.** Numerical data obtained during experiments represented in *Figure 6—figure supplement 4*.

and degradation via the UPS. Using MS and bioinformatics screening, we identified two E3 ligases, UBR5 and MARCHF7, that directly interact with nsp16 and mediate nsp16 ubiquitination and degradation (*Figure 8*).

UBR5, a member of the UBR box protein family, contains a HECT domain essential for its E3 ligase activity (*Kim et al., 2021*). UBR5 regulates tumor growth, metastasis (*Xiang et al., 2022*; *Qiao et al., 2020*), and viral infections. For instance, UBR5 targets the ORF4b protein of Middle East respiratory syndrome coronavirus for degradation, reducing its ability to counteract the host immune response (*Zhou et al., 2022*). Conversely, UBR5 supports Zika virus replication by assisting TER94/VCP in degrading the capsid protein, allowing viral genome release into the cytoplasm (Gestuveo et al.). MARCHF7, a RING E3 ligase, plays a role in T cell proliferation, neuronal development, inflammasome regulation, and tumor progression (*Zheng, 2021*; *Zhang et al., 2016*; *Cai et al., 2022*; *Zhao et al., 2018*). However, its involvement in viral infections remains largely unexplored, with one study identifying it as a host protein interacting with the 2C protein of foot-and-mouth disease virus (*Mahajan et al., 2021*).

Our findings demonstrate that UBR5 and MARCHF7 independently interact with nsp16 and facilitate its degradation, primarily on the cytoplasmic ER. Our findings highlight the ongoing evolutionary battle between viruses, which evolve strategies to evade host defenses, and hosts, which deploy multiple mechanisms to counteract viral infections. One such strategy is the use of multiple E3 ligases targeting the same viral protein for degradation. Functional analyses revealed that knocking down *UBR5* or *MARCHF7* enhanced SARS-CoV-2 replication, whereas their overexpression impaired it. Moreover, when their enzymatic activity was inactivated, the antiviral effect was abolished, confirming that their E3 ligase activity is essential for viral suppression. Notably, nsp16 overexpression alone did not increase viral levels in the absence of functional UBR5 or MARCHF7. This underscores the critical antiviral role of UBR5 and MARCHF7 in restricting SARS-CoV-2 by promoting nsp16 degradation.

Due to the presence of 18 lysine residues in nsp16, identifying specific ubiquitination sites was challenging. Using structural data (*Decroly et al., 2011*), we constructed truncated nsp16 mutants to assess their degradation profiles. Our results showed that nsp16 mutants degradation occurred to varying degrees (*Figure 8—figure supplement 1A and B*), suggesting that multiple lysine residues serve as recognition sites for E3 ligases. Furthermore, MG132 recovery experiments provided valuable insights. The Δ2–17 mutant, which lacks this segment, exhibited more than a twofold increase in

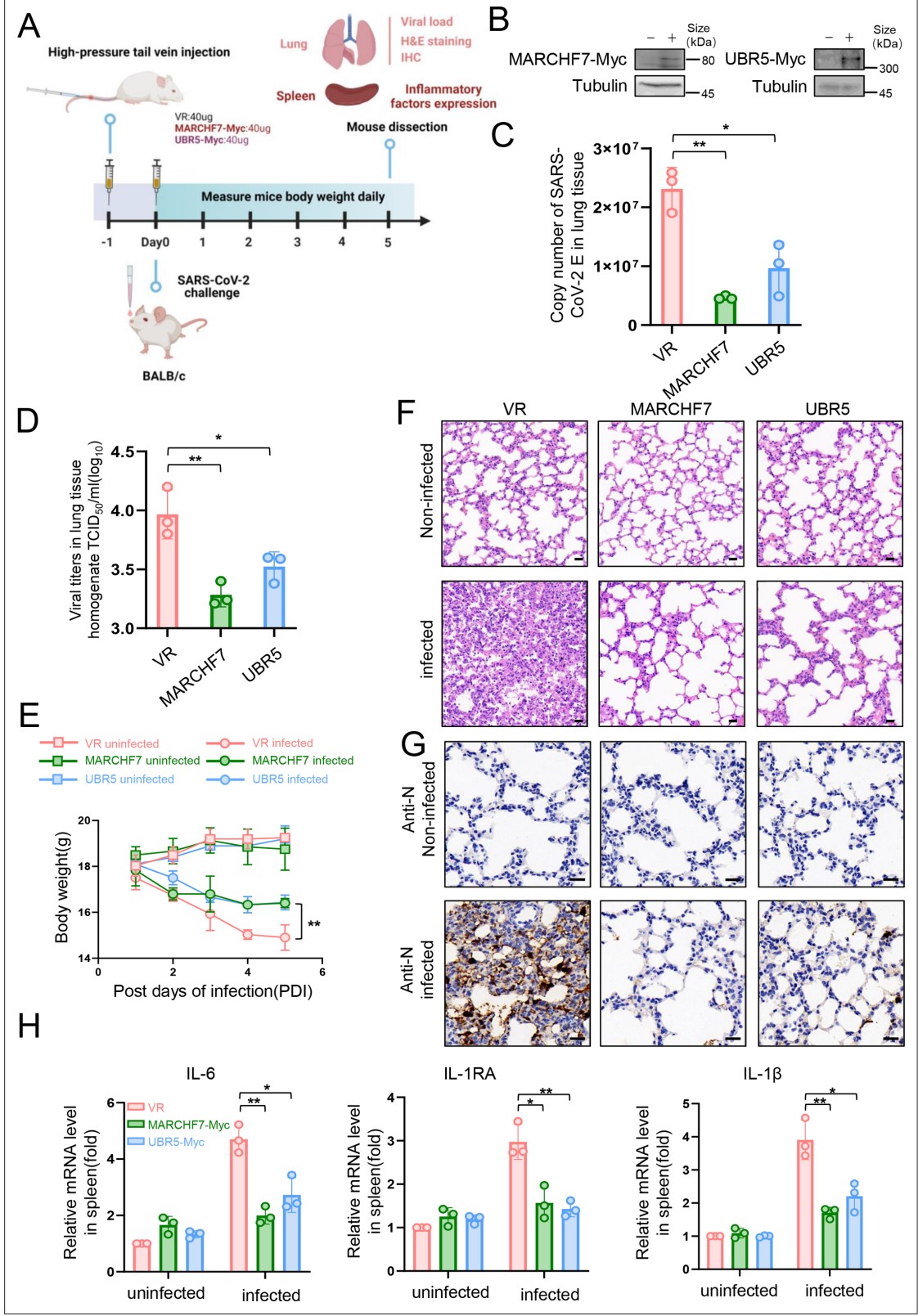

**Figure 7.** In a mouse infection model, overexpression of MARCHF7 or UBR5 exerted inhibitory effects on the virus. (**A–G**) BLAB/C mice were injected with the corresponding plasmids at 40 µg/500 µl via the high-pressure tail vein, followed by nasal inoculation with 50 µl severe acute respiratory syndrome coronavirus 2 (SARS-CoV-2) virus at a dosage of $10^{5.5}$ TCID50/ml (created with BioRender.com and the agreement no. is OO281XWHNA). Immunoblotting (IB) was used to detect the expression of MARCHF7 or UBR5 in the lung tissues (**B**). Viral RNA loads in mouse lung tissues were

*Figure 7 continued on next page*

*Figure 7 continued*

detected by measuring the mRNA levels of the E genes by RT-qPCR (**C**). Lung tissue was collected, homogenized, and the residue was removed by centrifugation to collect the supernatant. The viral titer was then measured using the TCID50 method (**D**). Mouse body weight was monitored during the experimental period (**E**). Representative images of hematoxylin and eosin (H&E) staining of lungs of mice with different treatments. Magnification, ×40. Scale bars, 20 µm (**F**). The staining of viral N proteins. Magnification, ×63. Scale bars, 20 µm. n=3 in each group (**G**). RT-qPCR was used to measure the expression of cytokines and chemokines in the spleens of mice in each group (**H**). Statistical significance was analyzed using a one-way analysis of variance with Tukey's multiple comparisons test (NS, no significance, *p<0.05, **p<0.01, ***p<0.001).

The online version of this article includes the following source data for figure 7:

**Source data 1.** PDF file containing original western blots for *Figure 7B*, indicating the relevant bands and treatments.

**Source data 2.** Original files for western blot analysis displayed in *Figure 7B*.

**Source data 3.** Numerical data obtained during experiments represented in *Figure 7*.

nsp16 expression, whereas the Δ204–212 mutant showed significantly reduced recovery after MG132 treatment compared to the wild-type. These observations offer important clues for further investigation into the interaction regions between nsp16 and MARCHF7 or UBR5, as well as potential ubiquitination sites. MS analysis identified K76 as a ubiquitination site (*Figure 8—figure supplement*

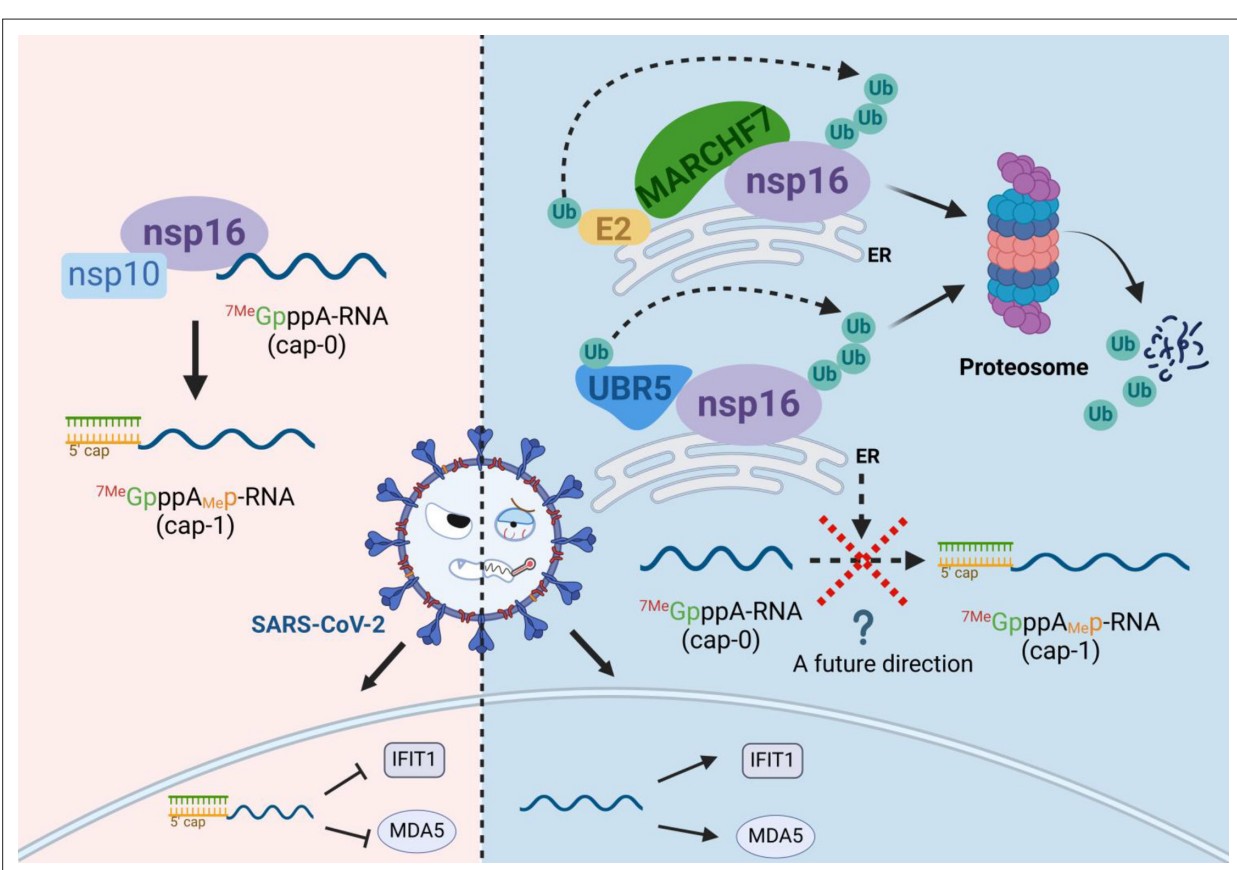

**Figure 8.** Schematic diagram of MARCHF7 and UBR5 ubiquitinate the severe acute respiratory syndrome coronavirus 2 (SARS-CoV-2) nonstructural protein nsp16, leading to its degradation via the proteasomal pathway, thereby affecting viral replication (created with BioRender.com and the agreement no. is EV281XWATL).

The online version of this article includes the following source data and figure supplement(s) for figure 8:

**Figure supplement 1.** Identification of the ubiquitination modification site of nsp16 protein.

**Figure supplement 1—source data 1.** PDF file containing original western blots for *Figure 8—figure supplement 1A, B, D, and E*, indicating the relevant bands and treatments.

**Figure supplement 1—source data 2.** Original files for western blot analysis displayed in *Figure 8—figure supplement 1A, B, D, and E*.

**Figure supplement 1—source data 3.** Numerical data obtained during experiments represented in *Figure 8—figure supplement 1*.

*1C*). However, the nsp16 K76R mutant still underwent degradation, though its ubiquitination level was reduced compared to that of the wild-type, indicating that K76 is one of several ubiquitination sites (*Figure 8—figure supplement 1D and E*). This suggests that nsp16 may undergo ubiquitination at multiple sites or through non-canonical pathways at non-lysine residues (*McClellan et al., 2019*). Beyond UBR5 and MARCHF7, other E3 ligases may also contribute to nsp16 regulation, warranting further investigation into additional host factors involved in SARS-CoV-2 restriction.

Currently, no specific activators or inhibitors exist for UBR5 or MARCHF7. To assess their antiviral effects in vivo, we used high-pressure tail vein injection, a gene delivery method primarily targeting the liver (*Kamimura et al., 2014*) but also effective for lung tissues (*Bonamassa et al., 2011*). Gene expression was successfully detected in the lungs, and while this method induced mild immune activation, its effect was minimal compared to other techniques (*Suda et al., 2023*). Notably, mice and nonhuman primates exhibit greater resistance to innate immune activation than humans (*Raper et al., 2003*), suggesting that immune activation had little influence on our results. However, translating these findings into therapeutic applications remains challenging. For instance, human trials using naked DNA injections (e.g. OTC cDNA) have shown variable immune responses, with one case of fatal immune activation reported (*Raper et al., 2003*). This underscores the variability of human immune responses and the challenges in translating findings from animal models to clinical applications. Such variability may also explain the inconsistent trends in MARCHF7 mRNA levels observed in peripheral blood samples from patients with different disease severities. Additionally, the small sample size in our study may have contributed to this observation.

In conclusion, in this study, we identified the host E3 ligases UBR5 and MARCHF7 as key regulators of SARS-CoV-2 nsp16, facilitating its ubiquitination and degradation. We observed a negative correlation between their expression levels and infection severity, highlighting their antiviral function. Moreover, both ligases effectively suppressed SARS-CoV-2 replication (*Figure 8*). These findings provide a foundation for developing UPS-targeted therapeutic strategies for COVID-19 treatment.

## Materials and methods

A list of materials is partially provided in *Supplementary file 1*.

### Reagents and antibodies

The drugs used in this study were as follows: MG132 (catalog no. S2619), Bafilomycin A1 (catalog no. S1413), Bortezomib (catalog no. S1013), Carfilzomib (catalog no. S2853), and Vinblastine (catalog no. S4504) were purchased from Selleck (Houston, TX, USA). Cycloheximide (catalog no. 66-81-9) was purchased from Sigma (St. Louis, MO, USA). The antibodies used for IB analysis were as follows: anti-UBR5 monoclonal antibody (mAb) (Proteintech, Rosemont, IL, USA, catalog no. 66937-1-Ig, RRID:AB_2881136), anti-MARCHF7 mAb (Santa Cruz Biotechnology, Dallas, TX, USA, catalog no. sc-166945, RRID:AB_2157974), SARS-CoV-2 2'-O-ribose Methyltransferase Antibody (nsp16) (Cell Signaling, Danvers, MA, USA, catalog no. #70811, RRID:AB_2799787), anti-Flag mAb (Sigma, catalog no. F1804, RRID:AB_262044), anti-tubulin mAb (Abcam, Cambridge, Cambridgeshire, UK, catalog no. ab11323, RRID:AB_297919), anti-hemagglutinin (anti-HA) pAb (Invitrogen, catalog no. 71-5500, RRID:AB_2533987), anti-Myc pAb (Proteintech, Rosemont, IL, USA, catalog no. 16286-1-AP, RRID:AB_2877901), and anti-GFP mAb (Abcam, catalog no. ab1218, RRID:AB_298079). Primary antibody: anti-COX5A pAb (Sangon Biotec, Shanghai, China, catalog no. D261450), anti-PDI-mAb (Proteintech, catalog no. 2E6A11, RRID:AB_2881137), and Human GM130/GOLGA2 Antibody (R&D Systems, Minneapolis, MN, USA, catalog no. AF8199, RRID:AB_2136393) were used for IF, and SARS-CoV-2 nucleocapsid mAb (GeneTex, Irvine, CA, USA, catalog no. GTX635679, RRID:AB_2909916) was used for immunohistochemistry. Fluorescent secondary antibodies: goat anti-Rabbit IgG (H+L) Highly Cross Adsorbed Secondary Antibody, Alexa Fluor Plus 488 (Invitrogen, catalog no. A11001, RRID:AB_2762824), and goat anti-Rabbit IgG (H+L) Highly Cross Adsorbed Secondary Antibody, Alexa Fluor Plus 568 (catalog no. A-11011, RRID:AB_143157) and immunohistochemical secondary antibodies: Streptavidin-Peroxidase Anti-Rabbit IgG kit (catalog no. KIT-9706) were purchased from Invitrogen and Maixin (Fuzhou, China) respectively.

## Cell lines and viruses

The cell lines used in this study include HEK293T cells (catalog no. CRL-11268), Hela cells (catalog no. CRM-CCL-2), Caco2 cells (catalog no. HTB-37). All cell lines were purchased from American Type Culture Collection (ATCC; Manassas, VA, USA) and tested for mycoplasma. In order to conduct virus-related experiments, stable cell lines including Caco2-N$^{int}$ (stably expressing SARS-CoV-2 N gene) and 293T-ACE2 (stably expressing ACE2) were generated in our laboratory. The cells were cultured in a cell incubator containing 5% $CO_2$ at 37°C using Dulbecco's modified Eagle's medium (HyClone, Logan, UT, USA) containing 10% heat-inactivated fetal calf serum (GIBCO BRL, Grand Island, NY, USA), 100 mg/ml penicillin, and 100 μg/ml streptomycin to provide nutrition. We expanded SARS-CoV-2 virus-like particles (GenBank access no. MN908947, Clade: 19A) with high replication capacity in Caco2-N$^{int}$ using a BSL-2 cell culture system. The virus infecting the cells includes Omicron BA.1 strain (human/CHN_CVRI-01/2022) and SARS-CoV-2 IME-BJ01 strain (BetaCoV/Beijing/IME-BJ05-2020, GenBank access no. MT291831.1), and mouse-adapted SARS-CoV-2/C57MA14 variant (GenBank: OL913104.1) used for experiments in mice was provided by Key Laboratory of Jilin Province for Zoonosis Prevention and Control, and all experiments for virus infection were performed in BSL-3 cell culture system.

## Plasmids

Eukaryotic expression plasmids encoding SARS-CoV-2 nsp protein were provided by Prof. Wang Peihui (Shandong University). The nsp16-HA expression vector was constructed by adding HA tag at the C-terminus. UBR5 (Gene ID: 51366) and its mutants (UBR5-ΔHECT, UBR5-ΔPABC, UBR5-ΔUBR) expression plasmids were constructed using purchased plasmids from Addgene (Watertown, MA, USA) as templates with no tag or a MYC tag at the N-terminus. The cDNA of 293T cells was used as a template to construct MARCHF7 (Gene ID: 64844) and its truncated mutant with a Myc tag at the C-terminus. All the above expression plasmids were inserted into the VR1012 vector. For IF and FRET experiments, pCDNA3.1-YFP vector (catalog no. 13033) and pECFP-C1 vector (catalog no. 6076-1) were purchased from Addgene and BD (Biosciences Clontech), and MARCHF7 or UBR5 was constructed on pECFP-C1, and nsp16 was constructed on pCDNA3.1-YFP. Single-point mutants of nsp16 of different virus subtypes were obtained by point mutagenesis. Multisite-combined mutants were synthesized by Sangon Biotec Company. Human Ub protein and its mutants have been previously described. Primers required for PCR were listed in *Supplementary file 1*.

## RNA extraction and real-time quantitative PCR

We used the method of TRIzol (Invitrogen, catalog no. 15596018CN) to extract RNA. The RNA was subsequently reverse-transcribed using a High-Capacity cDNA Reverse Transcription kit (Applied Biosystems, Carlsbad, CA, USA, catalog no. 4368814). Finally, PrimeScript RT Master Mix (Takara, Shiga, JPN, catalog no. RR036A) and relative real-time primers were used for qPCR on an Mx3005P instrument (Agilent Technologies, Stratagene, La Jolla, CA, USA). Amplification procedure of the target fragment is as follows: predenaturation at 95°C for 2 min, denaturation at 95°C for 30 s, annealing at 55°C for 30 s, and extension at 72°C for 30 s, total of 40 cycles. Real-time primer sequences used in this study were shown in *Supplementary file 1*.

## IB analysis

Cells or supernatant cultured for a certain time were collected, cell precipitates were resuspended with lysis buffer (50 mM Tris-HCl [pH 7.8], 150 mM NaCl, 1.0% NP-40, 5% glycerol, and 4 mM EDTA), 4× loading buffer was added (0.08 M Tris [pH 6.8], 2.0% SDS, 10% glycerol, 0.1 M dithiothreitol, and 0.2% bromophenol blue), and samples were lysed by heating at 100°C for 30 min. After removal of cell debris by centrifugation, the supernatant was taken to separate proteins of different sizes by SDS-PAGE. The proteins were transferred onto polyvinylidene fluoride membranes, incubated overnight with the indicated primary antibody, and after 1 hr at room temperature with HRP-conjugated secondary antibodies (Jackson ImmunoResearch, West Grove, NJ, USA, catalog no. 115-035-062 for anti-mouse and 111-035-045 for anti-rabbit), the proteins were visualized through Ultrasensitive ECL Chemiluminescence Detection Kit (Proteintech, catalog no. B500024).

## Stable cell line generation

We purchased the lentiviral vector pLKO.1-puro (catalog no. 8453) from Addgene. ShRNA targeting target genes was designed and synthesized by Sangon Biotech. After annealing, it was inserted into the vector. Lentivirus was generated by transfection with shRNA (cloned in pLKO.1) or control vector, RRE, VSVG, and REV, and the cells were screened with 1/1000 puromycin (3 µg/ml, Sigma, catalog no. P8833) 48 hr after infection. The knockdown efficiency was detected by RT-qPCR or IB. The shRNA target sequences used in this study are shown in *Supplementary file 1*.

## RNAi

siRNA used in this study was purchased from RiboBio Co. Ltd. (Guangzhou, China). The siRNA sequences for knockdown of *MARCHF7* or *UBR5* are provided in *Supplementary file 1*. The siRNA was transfected into the cells by transfection reagent Lipofectamine 2000 (Invitrogen, catalog no. 11668-019), and the corresponding plasmids were transfected 24 hr later using Lipofectamine 3000 Reagent (Invitrogen, Carlsbad, CA, USA, catalog no. L3000-008). The expression of target genes was detected by RT-qPCR or IB analysis 48 hr later.

## Immunoprecipitation

After 10 hr of MG132 (10 µM) treatment, the cells were harvested, resuspended in 1 ml lysis buffer (50 mM Tris-HCl [pH 7.5], 150 mM NaCl, 0.5% NP-40) containing protease inhibitor (Roche, catalog no. 11697498001), placed on the shaker oscillator for 4 hr to lyse, then centrifuged to remove cell debris, and the supernatant was incubated with the antibody and protein G agarose beads (Roche, Basel, Basel City, CH, catalog no. 11243233001) at 4°C overnight. Protein G agarose beads were washed six to eight times with wash buffer (20 mM Tris-HCl [pH 7.5], 100 mM NaCl, 0.1 mM EDTA, and 0.05% Tween 20) at 4°C, centrifuged at 800×*g* for 1 min. The proteins were eluted by adding loading buffer (0.08 M Tris [pH 6.8], 2.0% SDS, 10% glycerol, 0.1 M dithiothreitol, and 0.2% bromophenol blue), and the samples were boiled at 100°C for 10 min to elute the proteins. The lysates and immunoprecipitations were detected by IB.

## Mass spectrometry

HEK293T cells were transfected with SARS-CoV-2-nsp16-Flag for 36 hr and then harvested after 10 hr of treatment with MG132 or DMSO. Proteins were enriched by the Co-IP assay. And the elutions were analyzed by MS. MS analysis was performed by the National Center for Protein Science (Beijing, China).

## Immunofluorescence

48 hr after transfection, the solution was discarded, and the cells were washed twice with preheated PBS for 5 min each, followed by fixation with 4% paraformaldehyde at 37°C for 10 min. After three washes with PBS, permeabilized with 0.2% Triton X-100 for 5 min at 37°C, and then immediately followed by blocking with 10% fetal bovine serum at room temperature for 1 hr. The blocked cells were incubated overnight with the indicated primary antibodies. The next day, after three washes with PBS, the cells were incubated with corresponding secondary antibodies for 1 hr at room temperature in the dark. The nuclei were stained with DAPI (4',6-diamidino-2-phenylindole, Sigma, catalog no. 9542) and then stored in 90% glycerol. The fluorescence was detected by a laser scanning confocal microscope (FV 3000, Olympus, Tokyo, Japan).

## Viral infectivity assay

The Caco2 cells were transfected with siRNA by Lipofectamine RNAiMAX Reagent (Invitrogen, catalog no. 13778150) to knock down *UBR5* or *MARCHF7* and infected with IME-BJ01 strain (MOI = 0.01) or Omicron-BA.1 strain (MOI = 0.001) 24 hr after transfection. After 2 hr of infection, the Caco2 cells were cultured in fresh medium containing 2% fetal bovine serum for 48 hr. The cells and supernatants were collected. Viral levels were characterized by RT-qPCR and IB analysis of viral structural proteins. The overexpression plasmids UBR5-Myc, MARCHF7-Myc, and nsp16-Flag were transfected in 293T-ACE2 by Lipofectamine 3000 Reagent and infected with virus 24 hr later.

## Mouse lines and infections

BALB/C mice, 6 weeks, were purchased from Charles River Laboratories, Beijing. Mice were cultured in groups according to experimental groups. Our study examined male and female animals, and similar findings are reported for both sexes. To reduce animal suffering, all welfare and experimental procedures were carried out in strict accordance with the Guide for the Care and Use of Laboratory Animals and relevant ethical regulations during the experiment. The mice were randomly divided into six groups and injected with 40 µg/500 µl MARCHF7 or UBR5 plasmids via the tail vein at high pressure. Three groups of mice (MARCHF7-injected, UBR5-injected, and empty vector VR1012-injected) were infected 50 µl with SARS-CoV-2 isolates by nasal challenge at a dose of $10^{5.5}$ $TCID_{50}$/ml. Three groups were not infected with virus as control, and the empty vector injection group was also used as control.

## Immunohistochemical analysis

Three mice in each group, a total of 18 mice, were sacrificed after anesthesia. The lungs of mice were collected and fixed in 4% paraformaldehyde fixative. After 2 days, the lungs were progressively dehydrated through different concentrations of ethanol solution, transparently treated, soaked in xylene, and finally embedded in paraffin. Some thin sections obtained through a microtome for hematoxylin and eosin staining. The other fraction was incubated with 3% hydrogen peroxide for 5–10 min at room temperature to eliminate endogenous peroxidase activity. SARS-CoV-2 nucleocapsid mAb (GeneTex, catalog no. GTX635679) and a Streptavidin-Peroxidase Anti-Rabbit IgG kit (Maixin, Fuzhou, China, catalog no. KIT-9706) were used to quantify the viral level in lung tissue.

## Statistical analysis

The statistical analyses used in the figures have been described in detail in the figure legends. All data were expressed as mean ± standard deviations (SDs). Statistical comparisons were performed using Student's t-test, one-way ANOVA, or repeated measures ANOVA. The differences were statistically significant as follows: $*p<0.05$, $**p<0.01$, $***p<0.001$; ns is for no meaning.

## Acknowledgements

We thank CY Dai for providing critical reagents. This work was supported by funding from the National Natural Science Foundation of China (82341072 and 82272316 to ZWY, 82341062 to LZL), the National Key R&D Program of China (2021YFC2301900 and 2301904, 2023YFC2306603), the Science and Technology Department of Jilin Province (YDZJ202301ZYTS521 and YDZJ202201ZYTS587), the Key Laboratory of Molecular Virology, Jilin Province (20102209), and Bethune Project, Jilin University (2023B03). The funding sources were involved in study design, data collection, and interpretation, and the decision to submit the work for publication.

## Additional information

### Funding

| Funder | Grant reference number | Author |
| --- | --- | --- |
| National Natural Science Foundation of China | 82341072 | Wenyan Zhang |
| National Natural Science Foundation of China | 82272316 | Wenyan Zhang |
| National Natural Science Foundation of China | 82341062 | Zhaolong Li |
| National Key Research and Development Program of China | 2021YFC2301900 and 2301904 | Wenyan Zhang |
| National Key Research and Development Program of China | 2023YFC2306603 | Wenyan Zhang |

| Funder | Grant reference number | Author |
|---|---|---|
| Science and Technology Department of Jilin Province | YDZJ202301ZYTS521 | Wenyan Zhang |
| Science and Technology Department of Jilin Province | YDZJ202201ZYTS587 | Zhaolong Li |
| Key Laboratory of Molecular Virology, Jilin Province | 20102209 | Wenyan Zhang |
| Jilin University | 2023B03 | Zhaolong Li |

The funders had no role in study design, data collection and interpretation, or the decision to submit the work for publication.

## Author contributions

Li Tian, Conceptualization, Data curation, Formal analysis, Investigation, Writing – original draft, Project administration, Writing – review and editing; Zongzheng Zhao, Data curation, Methodology, Writing – original draft; Wenying Gao, Zirui Liu, Xiao Li, Resources, Methodology; Wenyan Zhang, Conceptualization, Resources, Formal analysis, Funding acquisition, Methodology, Project administration, Writing – review and editing; Zhaolong Li, Conceptualization, Data curation, Formal analysis, Funding acquisition, Investigation, Methodology, Writing – review and editing

## Author ORCIDs

Li Tian ⓘ https://orcid.org/0009-0007-6074-4940
Wenyan Zhang ⓘ https://orcid.org/0000-0003-4507-521X
Zhaolong Li ⓘ https://orcid.org/0000-0003-3795-7220

## Ethics

Collection of inpatient blood was approved by the Ethics Committee of the First Hospital of Jilin University (21K105-001) in accordance with the guidelines and principles of The World Medical Association (WMA) Declaration of Helsinki and the Department of Health and Human Services Belmont Report. All study participants signed an informed consent form. PBMCs were obtained from nine COVID-19 patients with mild disease, six patients with severe disease, and five patients with critical disease (Supplementary file 2).

All animal experiments were approved by the ethics committee of the Research Unit of Key Technologies for Prevention and Control of Virus Zoonoses, Chinese Academy of Medical Sciences, Changchun Veterinary Research Institute, Chinese Academy of Agricultural Sciences (IACUC of AMMS-11-2020-006).

Reviewer #1 (Public review): https://doi.org/10.7554/eLife.102277.4.sa1
Reviewer #3 (Public review): https://doi.org/10.7554/eLife.102277.4.sa2
Author response https://doi.org/10.7554/eLife.102277.4.sa3

# Additional files

## Supplementary files

Supplementary file 1. Primers used in this study.

Supplementary file 2. Patients' clinical information in this study.

MDAR checklist

## Data availability

The authors declare that all data supporting the findings of this study are available within the paper and its supplementary files. The mass spectrometry proteomics data have been deposited to the ProteomeXchange Consortium (http://proteomecentral.proteomexchange.org) via the iProX partner repository with the dataset identified PXD053961.

The following dataset was generated:

| Author(s) | Year | Dataset title | Dataset URL | Database and Identifier |
|---|---|---|---|---|
| Zhang W | 2024 | Proteins interacting with SARS-CoV-2 NSP16 | https://proteomecentral.proteomexchange.org/cgi/GetDataset?ID=PXD053961 | ProteomeXchange, PXD053961 |

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
