## [Editor Report · eLife Assessment]

This **important** work advances our understanding of how the SARS-CoV-2 Nsp16 protein is regulated by host E3 ligases to promote viral mRNA capping. Support for the overall claims in the revised manuscript is **convincing** . This work will be of interest to those working in host-viral interactions and the role of the ubiquitin-proteasome system in viral replication.

---

## [Referee Report · Reviewer #1 (Public review)]

In this study, Tiang et al. explore the role of ubiquitination of non-structural protein 16 (nsp16) in the SARS-CoV-2 life cycle. nsp16, in conjunction with nsp10, performs the final step of viral mRNA capping through its 2'-O-methylase activity. This modification allows the virus to evade host immune responses and protects its mRNA from degradation. The authors demonstrate that nsp16 undergoes ubiquitination and subsequent degradation by the host E3 ubiquitin ligases UBR5 and MARCHF7 via the ubiquitin-proteasome system (UPS). Specifically, UBR5 and MARCHF7 mediate nsp16 degradation through K48- and K27-linked ubiquitination, respectively. Notably, degradation of nsp16 by either UBR5 or MARCHF7 operates independently, with both mechanisms effectively inhibiting SARS-CoV-2 replication in vitro and in vivo. Furthermore, UBR5 and MARCHF7 exhibit broad-spectrum antiviral activity by targeting nsp16 variants from various SARS-CoV-2 strains. This research advances our understanding of how nsp16 ubiquitination impacts viral replication and highlights potential targets for developing broadly effective antiviral therapies.

Strengths:

The proposed study is of significant interest to the virology community because it aims to elucidate the biological role of ubiquitination in coronavirus proteins and its impact on the viral life cycle. Understanding these mechanisms will address broadly applicable questions about coronavirus biology and enhance our overall knowledge of ubiquitination's diverse functions in cell biology. Employing in vivo studies is a strength.

Weaknesses:

Minor comments:

Figure 5A- The authors should ensure that the figure is properly labeled to clearly distinguish between the IP (Immunoprecipitation) panel and the input panel.

---

## [Referee Report · Reviewer #3 (Public review)]

Summary:

The manuscript "SARS-CoV-2 nsp16 is regulated by host E3 ubiquitin ligases, UBR5 and MARCHF7" is an interesting work by Tian et al. describing the degradation/ stability of NSP16 of SARS CoV2 via K48 and K27-linked Ubiquitination and proteasomal degradation. The authors have demonstrated that UBR5 and MARCHF7, an E3 ubiquitin ligase bring about the ubiquitination of NSP16. The concept, and experimental approach to prove the hypothesis looks ok. The in vivo data looks ok with the controls. Overall, the manuscript is good.

Strengths:

The study identified important E3 ligases (MARCHF7 and UBR5) that can ubiquitinate NSP16, an important viral factor.

Comments on revisions:

I had gone through the revised form of the manuscript thoroughly. The authors have addressed all of my concerns. To me, the experimental approach looks convincing that the host E3 ubiquitin ligases (UBR5 and MARCHF7) ubiquitinate NSP16 and mark it for proteasomal degradation via K48- and K27- linkage. The authors have represented the final figure (Fig.8) in a convincing manner, opening a new window to explore the mechanism of capping the vRNA bu NSP16.

---

## [Author Response]

The following is the authors’ response to the previous reviews

**Public Reviews:**

**Reviewer #1 (Public review):**
In this study, Tian et al. explore the role of ubiquitination of non-structural protein 16 (nsp16) in the SARS-CoV-2 life cycle. nsp16, in conjunction with nsp10, performs the final step of viral mRNA capping through its 2'-O-methylase activity. This modification allows the virus to evade host immune responses and protects its mRNA from degradation. The authors demonstrate that nsp16 undergoes ubiquitination and subsequent degradation by the host E3 ubiquitin ligases UBR5 and MARCHF7 via the ubiquitin-proteasome system (UPS). Specifically, UBR5 and MARCHF7 mediate nsp16 degradation through K48- and K27-linked ubiquitination, respectively. Notably, degradation of nsp16 by either UBR5 or MARCHF7 operates independently, with both mechanisms effectively inhibiting SARS-CoV-2 replication in vitro and in vivo. Furthermore, UBR5 and MARCHF7 exhibit broad-spectrum antiviral activity by targeting nsp16 variants from various SARS-CoV-2 strains. This research advances our understanding of how nsp16 ubiquitination impacts viral replication and highlights potential targets for developing broadly effective antiviral therapies.Strengths:The proposed study is of significant interest to the virology community because it aims to elucidate the biological role of ubiquitination in coronavirus proteins and its impact on the viral life cycle. Understanding these mechanisms will address broadly applicable questions about coronavirus biology and enhance our overall knowledge of ubiquitination's diverse functions in cell biology. Employing in vivo studies is a strength.Weaknesses:Minor comments:Figure 5A- The authors should ensure that the figure is properly labeled to clearly distinguish between the IP (Immunoprecipitation) panel and the input panel.

Thank you for your suggestion. We have exchanged Figure 5 in this version.

**Reviewer #3 (Public review):**
Summary:The manuscript "SARS-CoV-2 nsp16 is regulated by host E3 ubiquitin ligases, UBR5 and MARCHF7" is an interesting work by Tian et al. describing the degradation/ stability of NSP16 of SARS CoV2 via K48 and K27-linked Ubiquitination and proteasomal degradation. The authors have demonstrated that UBR5 and MARCHF7, an E3 ubiquitin ligase bring about the ubiquitination of NSP16. The concept, and experimental approach to prove the hypothesis looks ok. The in vivo data looks ok with the controls. Overall, the manuscript is good.Strengths:The study identified important E3 ligases (MARCHF7 and UBR5) that can ubiquitinate NSP16, an important viral factor.Comments on revisions:I had gone through the revised form of the manuscript thoroughly. The authors have addressed all of my concerns. To me, the experimental approach looks convincing that the host E3 ubiquitin ligases (UBR5 and MARCHF7) ubiquitinate NSP16 and mark it for proteasomal degradation via K48- and K27- linkage. The authors have represented the final figure (Fig.8) in a convincing manner, opening a new window to explore the mechanism of capping the vRNA bu NSP16.

Thank you for your recognition.